# The Involvement of the Multiple Demand and Default Mode Networks in a Trial-by-Trial Cognitive Control

**DOI:** 10.3390/brainsci13091247

**Published:** 2023-08-26

**Authors:** Shinyoung Jung, Joo Yeon Kim, Suhyeon Jo, Suk Won Han

**Affiliations:** 1Department of Psychological Sciences, Texas Tech University, MS 2051, Lubbock, TX 79409, USA; shinyoung.jung@ttu.edu; 2Department of Research Equipment Operation, Korea Basic Science Institute, Cheong-won, Ochang 28119, Republic of Korea; jooyun8992@kbsi.re.kr; 3Department of Psychology, Chungnam National University, Daejeon 34134, Republic of Korea; j77460744@gmail.com

**Keywords:** fMRI, multiple demand network, default mode network, task coding, MVPA

## Abstract

Adaptive behavior in the environment requires a high level of cognitive control to bias limited processing resources to behaviorally significant stimuli. Such control has been associated with a set of brain regions located in the fronto-parietal cortex (multiple demand network), whose activity was found to increase as the control demand for a task increases. In contrast, another set of regions, default mode network regions, were found to be deactivated during top-down processing of task stimuli. Despite this dissociation in their activation amplitudes, it is possible that activation patterns of these regions commonly encode specific task features. In two independent neuroimaging datasets, involving a total of 40 human samples, we found that the performance of an attentional task evoked positive activity of the MDN and deactivation of the DMN. Consistent with previous studies, task features could be decoded from the fronto-parietal cognitive regions. Importantly, the regions of the DMN also encoded task features when the task set had to be rapidly reconfigured in a transient, trial-by-trial manner, along with the MDN regions. These results suggest that the two separate brain networks ultimately co-ordinate for the effective establishment of top-down cognitive control.

## 1. Introduction

The performance of a cognitive task requires the formulation of a task set, biasing limited processing resources toward task-related stimuli in a top-down manner. Such control has been associated with a network of brain regions on the fronto-parietal cortex, including lateral prefrontal cortex (lPFC), anterior insula (AI), anterior cingulate cortex (ACC), and intraparietal sulcus (IPS) [1,2,3,4]. These regions are known to be parts of the multiple demand network (MDN), commonly recruited by a wide range of cognitive tasks; activation amplitudes of these regions increased as the demands of a variety of tasks increased [2,3,5]. Furthermore, recent studies showed that specific task features could be decoded from activation patterns of the MDN. Specifically, in previous studies, in the face of a multi-dimensional target stimulus, researchers were able to infer which stimulus dimension was relevant and attended by examining activation patterns of the MDN regions [4,6].

Besides the multiple demand network, another functionally significant brain network has been identified [7,8]. A set of regions including the posterior cingulate cortex (PCC), temporal-parietal junction (TPJ), and medial prefrontal cortex (mPFC) was found to form a default mode network (DMN). In contrast to the MDN, whose activity increases as task demand increases, the DMN regions were found to be deactivated with increasing task demands [5,8,9,10,11]. While the MDN has been implicated in top-down control of attention, working memory, and other elementary cognitive processes, the functional role of the DMN has been associated with higher level reasoning and social cognitive processing, such as semantic processing, self-referential processing, and inferring others’ mental states [6,12,13,14,15].

As briefly reviewed above, the MDN and DMN have been recognized to play differential or even opposing roles. However, recent evidence shows that these two networks ultimately co-ordinate for optimal task performance. Specifically, Wen and colleagues demonstrated that both the DMN and MDN contributed to the performance of a cognitive task, with the DMN representing a relatively broad and long-term task context and the MDN encoding specific details regarding each individual task feature and response [6]. Indeed, an earlier study also showed that a contextual change was an important factor to activate the DMN [16].

Taken together, based upon extant evidence, it seems plausible that the DMN represents a long-term task context and contextual changes evoking cognitive set switches. A remaining unresolved issue is whether each individual region of the DMN also plays a role in a transient, trial-by-trial adjustment of cognitive control. While such control has been primarily associated with the fronto-parietal or MDN regions, the contribution of the DMN to this kind of control is unknown. Indeed, growing evidence shows that some brain regions encoding task features overlap with the DMN regions [17,18,19]. However, it was not verified that those regions encoding task features show a typical characteristic of the DMN, deactivation during task performance. In the present study, we examined both activation amplitudes and activation patterns to directly test whether deactivated regions during an attention-demanding task would also encode task features, along with the task-positive regions, such as the MDN regions.

To address this issue, we employed a well-established, cued-task switching paradigm [19]. In this paradigm, participants face a multi-dimensional task stimulus. There are multiple rules to correctly respond to the target stimulus. Prior to the target presentation, a cue is presented to inform participants which dimension should be attended and which task rule should be applied. Numerous previous studies showed that activation patterns of the MDN or the fronto-parietal regions discriminated which task rule was applied [4,17,20,21]. Here, we investigated whether the DMN also encodes such task features in a condition, under which an attentional set should be formulated and adjusted in a trial-by-trial manner. In two independent sets of functional magnetic resonance imaging (fMRI) data, involving a total of 40 human samples, we demonstrated that the performance of an attentional task evoked positive activity of the MDN and deactivation of the DMN. Importantly, when the cue–target SOA was relatively short, some regions of the DMN were found to encode task features, along with the MDN regions.

## 2. Methods

### 2.1. Participants

A total of 40 adults (16 males, aged 18–30) who had normal (or corrected to normal) vision voluntarily participated and received monetary compensation. All the participants reported no issue in performing the task. To determine the sample size, we considered previous studies, including our own study, employing similar paradigm and analyses [17,18,19]. Sample sizes of these studies ranged from 14 to 40. Given the issue of reproducibility of fMRI data, we decided to collect data from two independent sets of samples, each of which included 20 participants. The experimental protocol was approved by the Chungnam National University Institutional Review Board and written informed consent was obtained from each participant.

### 2.2. Stimuli and Apparatus

Identical stimuli and apparatus were used for Experiment 1 and Experiment 2. The experiment was programmed and run using Psychopy2 (Peirce, 2007). The stimuli were projected to a mirror that was mounted onto the head coil in the scanner. Experimental stimuli included cue words and target images. The cue stimuli were “얼굴” or “장소”, which were Korean words meaning face or place, respectively. The height of each cue word was 2° of visual angle. Target stimuli were created by superimposing two images from different categories (face and building), which subtended 6° × 6° of visual angle. Face images were taken from the FEI Face Database [22] and University of Texas at Dallas face image database [23]. Building images were taken from SUN image database [24].

### 2.3. Design and Procedure

Participants were instructed to perform a cued switching task. The target stimuli were superimposed images of a face and a building. Participants were required to either judge the gender of the face or identify the building structure (1 story or 2 story). At the beginning of each trial, a task cue was presented at the center of display for 2000 ms. These cues instructed participants to perform the face or place task (Figure 1). The cue presentation was followed by a blank interval of variable durations. In Experiment 1, the duration of this cue–target interval followed an exponential distribution (16 trials × 4 s, 8 trials × 8 s, 4 trials × 12 s). Then, the target was presented for 150 ms, followed by a blank inter-trial interval that also followed an exponential distribution (16 trials × 4 s, 8 trials × 8 s, 4 trials × 12 s). There were six functional MRI run, each of which included 28 trials. All the details of the Experiment 2 were identical to those of Experiment 1, except that the cue–target interval was chosen from the values of 500 ms (7 trials), 1500 ms (7 trials), 2500 ms (7 trials), or 7500 ms (7 trials). These values were chosen based upon previous studies [17,19]. Due to this change, the cue duration was shortened to 500 ms.

### 2.4. fMRI Methods

For each participant, an anatomical T1-weighted whole-brain image was acquired with conventional parameters on Philips Achieva 3T scanner at the Korea Basic Science Institute (FOV: 256 × 256, slice thickness: 1 mm with no gap, in plane resolution: 1 mm × 1 mm). For functional magnetic resonance image scan, a total of 272 brain volumes were obtained in each run (T2*-weighted). Thirty-three 3.5 axial slices (0.5 mm skip; 3.75 × 3.75 mm in-plane) were taken parallel to the AC-PC line (TR, 2000 ms; TE, 35 ms; FA, 79°; FOV, 240 mm). The imaging data were analyzed by using the FMRIB Software Library (FSL, http://fsl.fmrib.ox.ac.uk).

Preprocessing of the imaging data included motion correction (MCFLIRT), brain image acquired time correction, brain skull removal, spatial smoothing (5 mm Gaussian kernel), and high-pass filtering (100 Hz). All functional data were co-registered to each individual’s T1-weighted anatomical brain image and were normalized to Montreal Neurological Institute space brain template.

To define regions of interest (ROIs), a conventional univariate generalized linear model analysis (GLM) was performed. Specifically, four regressors (two types of cues and targets: face-cue, place-cue, face-target, and place-target) were created by using the volumes corresponding to the cue and target onset, which were convolved with a double-gamma hemodynamic response function (HRF). Then, a contrast was run to identify brain regions that were significantly activated (activation contrast) either by the cue or the target presentation; regression coefficients of 1 were assigned to all regressors. A second contrast (deactivated contrast), in which regression coefficients of −1 were assigned to four regressors, was also run to identify brain regions that were deactivated by the presentation of the cue or the target. The resulting group statistical parametric maps (SPM) were cluster-thresholded (Z = 2.3, corrected *p* < 0.05). The ROIs were created by drawing a sphere whose radius was 8 mm, centering on peak voxels of each significant activational foci. Since no hemispheric difference was found, data from bilateral ROis were collapsed. All the participants’ brain image data from Experiment 1 and Experiment 2 were included in this group-level analysis to define consistent ROIs across the two experiments. The co-ordinates of the ROIs are described in Table 1.

We also employed multivariate pattern analyses (MVPA) of the brain responses data to investigate whether the ROIs would show differential pattern of activation, depending on task types [4,17,25,26]. Specifically, we examined whether the cue-evoked activation and task-evoked activation would discriminate the task types. For the MVPA, the preprocessed fMRI data (see above for details) underwent Z-transformation and polynomial trend removal. Further processing steps were applied to these data when needed. Specifically, in the case that significant univariate difference was found across the task conditions, we subtracted the mean activity of all the voxels for each condition from each voxel activity. Then, we reapplied the MVPA to these mean-centered data. None of the MVPA results were affected by this procedure. Furthermore, since the reaction time (RT) varied across the task conditions (face vs. place), we regressed out the variance associated with RT when applying the MVPA to the target-evoked activity data. This process did not affect the main results either [27].

From a total of six runs, all but two functional runs of brain image data were used to train a linear support vector machine. Then, the data of the held-out runs were tested on the trained classifier (two-fold cross-validation). This process was repeated until all the runs served as test runs. To create event-related MVPA plots, this decoding analysis was conducted for each ROI and TR separately. This MVPA procedure allows one to verify that significant decoding is driven by genuine hemodynamic responses [28]. Each ROI’s decoding performance was assessed via one-sample *t*-tests, comparing the peak decoding accuracy against the chance level performance (50%). The statistical thresholds of these analyses were Bonferroni-corrected by dividing the *p*-values by the number of total ROIs (7).

Finally, although trials randomly varied in terms of task switch versus task repeat (relative to the prior trial), we intentionally collapsed across this variable [17]. This was because we were primarily interested in task coding, rather than task switching. The present approach is also optimal to maximize the number of trials for the decoding analysis.

## 3. Results

### 3.1. Behavioral Results

Target accuracy for the face task was significantly greater than for the place task, t(19) = 7.349, *p* < 0.001, d = 1.907. The reaction time (RT) metrics aligned with this accuracy trend; the RT for the face task (M = 955 ms) was significantly shorter than the RT for the place task (M = 1206 ms), t(19) = 5.325, *p* < 0.001, Cohen’s d = 0.939.

In Experiment 2, we found no significant difference across the task types in accuracy, t(19) = 2.008, *p* = 0.059, d = 0.574. The RT data showed a similar pattern with that of Experiment 1. The face task (M = 970 ms) yielded significantly shorter response times relative to the place task (M = 1065 ms), t(19) = 3.148, *p* = 0.005, d = 0.343 (see Figure 2C,D).

### 3.2. Univariate GLM Results

A whole-brain univariate analysis of fMRI data revealed that a large set of brain regions were significantly activated either by the cue presentation or by the target presentation. These regions included the dorsal anterior cingulate cortex (dACC), lateral prefrontal cortex (LPFC), anterior insula (AI), and intraparietal sulcus (IPS). These regions corresponded well to the MDN regions identified in previous studies and served as our MDN ROIs [4,5,29]. We ran another whole-brain univariate analysis to define the DMN ROIs (see the Section 2 for details). These included bilateral temporal–parietal junction (TPJ), posterior cingulate cortex (PCC), and medial prefrontal cortex (mPFC). Since no hemispheric difference was found, we collapsed bilateral ROIs, yielding a total of seven ROIs.

After identifying regions of interest (ROIs) using whole-brain SPM analyses, we analyzed the BOLD activation amplitudes of the ROIs across the task types. In Experiment 1, there were no significant differences in cue activations across task conditions. We observed difference in task-evoked activations across task conditions in one of the MDN regions, the IPS (Figure 3). This region showed significantly greater activity when participants performed the place task when they performed the face task, t(19) = 3.113, *p* < 0.040, d = 0.603, Bonferroni-corrected. Considering that participants’ responses were slower and less accurate for the place task than for the face task in Experiment 1, this finding supports the idea that the IPS is engaged when attention resources are in high demand.

Regarding the ROIs in the DMN, target presentation evoked deactivation of these regions. Furthermore, there were significant activation differences across the task conditions in the TPJ, t(19) = 3.040, *p* < 0.047, d = 0.704, Bonferroni-corrected, and in the pCC t(19) = 3.317, *p* < 0.025, d = 0.865. This corresponds to the traditional theory of the default mode network, suggesting that the DMN regions were deactivated when tasks require significant attention, as the place task was more challenging for participants based on their behavioral results.

In Experiment 2 (Figure 4), we found significant differences in amplitude of cue-evoked activity in the LPFC across the task conditions, t(19) = 3.464, *p* < 0.018, d = 0.473, Bonferroni-corrected. The IPS activity showed significant differences across the task conditions both during the cue presentation, t(19) = 3.444, *p* < 0.019, d = 0.484, and during the target presentation, t(19) = 3.266, *p* < 0.028, d = 0.590, Bonferroni-corrected (Figure 4). In contrast, deactivations were observed in the DMN brain areas during target presentation. Furthermore, the mPFC activity showed significant difference in activation levels across task conditions during target presentation, t(19) = 3.818, *p* = 0.008, d = 0.837, Bonferroni-corrected.

Taken together, across two experiments, we found that the MDN regions were activated during task performance, whereas the DMN regions were deactivated. Having confirmed that our MDN and DMN showed typical characteristics in terms of activation amplitudes, we turn to multivariate pattern analyses of these regions’ data.

### 3.3. Event-Related Multivariate Pattern Analysis

A primary reason for utilizing the MVPA was to investigate whether the regions deactivated during the performance of the task would also encode task features, just as the activated regions do. To preview the results, across the two experiments, the target-evoked activation patterns in the LPFC and IPS consistently discriminated which task rule (Face or Place task) was applied. Such a trend was also found in the TPJ and mPFC. Regarding the cue-evoked activation pattern data, no evidence for task decoding was found in Experiment 1, in which long cue–target SOAs were used (see Methods for details). However, in Experiment 2, in which relatively short cue–target SOAs were used, cue-evoked activation patterns in some regions of the MDN and DMN discriminated the task rules. Decoding accuracies based upon target-evoked activation generally enhanced in Experiment 2, and target-activation patterns in all the ROIs of the MDN and DMN, except for the AI, discriminated task rules.

In Experiment 1, the MVPA revealed that there were no cue-specific (face vs. place) multi-voxel patterns in all the ROIs of the MDN and DMN. However, the LPFC (t(19) = 3.698, *p* = 0.010, d = 0.826, Bonferroni-corrected), and IPS (t(19) = 5.673, *p* = 0.0001, d = 1.268, Bonferroni-corrected) of the MDN coded task-specific representations during the target presentation (Figure 5). We also found that the TPJ and mPFC activation patterns evoked by the target presentation discriminated the task rules; peak decoding accuracies of these regions (about 52%) were significantly above the chance level, but these did not survive the Bonferroni correction for multiple comparisons. Notably, similar decoding accuracies from these regions were also reported in a previous study [19].

From the MVPA results of Experiment 2 (Figure 6), we found significant cue and target decoding in some ROIs. Specifically, the cue-evoked activation patterns of the dACC (t(19) = 3.853, *p* = 0.007, d = 0.861, Bonferroni-corrected), LPFC (t(19) = 5.336, *p* = 0.0002, d = 1.193, Bonferroni-corrected), and IPS (t(19) = 4.130, *p* = 0.003, d = 0.923, Bonferroni-corrected) discriminated the task rules. Significant cue decoding was also found in the data of TPJ (t(19) = 4.721, *p* = 0.001, d = 1.055, Bonferroni-corrected) and pCC (t(19) = 3.121, *p* = 0.039, d = 0.698, Bonferroni-corrected). These cue decoding accuracies were greater for Experiment 2 than for Experiment 1; peak decoding accuracies of these regions of the two experiments were compared via independent sample *t*-tests, *p* < 0.005. The anterior insula of MDN and mPFC of DMN did not have cue-specific representations according to the MVPA results.

Regarding the target-evoked activation pattern data, the activation patterns in the dACC (t(19) = 3.419, *p* = 0.020, d = 0.764, Bonferroni-corrected), LPFC (t(19) = 4.114, *p* = 0.004, d = 0.920, Bonferroni-corrected), and IPS (t(19) = 5.546, *p* = 0.0001, d = 1.240, Bonferroni-corrected) discriminated task rules. The DMN regions, the mPFC (t(19) = 4.379, *p* = 0.002, d = 0.979, Bonferroni-corrected), TPJ (t(19) = 3.555, *p* = 0.014, d = 0.794, Bonferroni-corrected), and pCC (t(19) = 4.297, *p* = 0.002, d = 0.961, Bonferroni-corrected) also encoded task rules (face vs. task). Target decoding accuracies of these DMN regions were significantly greater for Experiment 2 than for Experiment 1, *p* < 0.005.

To summarize the MVPA results, in Experiment 1, in which relatively long cue–target intervals were used, the LPFC and IPS in the MDN encoded task rules during the target presentation. The target-evoked activation patterns in the mPFC and TPJ also seemed to discriminate task rules, but their decoding performance was not significant above the multiple comparison-corrected statistical thresholds. We found no evidence that cue-evoked activation patterns encoded task features.

However, in Experiment 2 with short cue–target intervals, significant cue decoding was found; cue-evoked activation patterns in the dACC, LPFC, and IPS of the MDN discriminated task rules, as did the patterns in TPJ and pCC of the DMN. Regarding the target-evoked activation patterns, it was found that all the ROIs’ activation patterns, except for that of the AI, discriminated task rules. Finally, the overall decoding performance was greater for Experiment 2 than for Experiment 1. The enhance decoding performance of Experiment 2, compared to that of Experiment 1, is discussed in the Section 4.

## 4. Discussion

In this study, we examined how the MDN and DMN contribute to a trial-by-trial adjustment of cognitive task sets. Specifically, we examined activity of these regions when participants were engaged with the process of configuring a task set and applying the appropriate task rule in the face of a multi-dimensional target stimulus. In this series of processes, the MDN regions were activated when either the task cue or the target was presented, whereas the DMN regions were deactivated during target processing. While the univariate activities of these two networks were contrasting, it could be decoded from activation patterns of several MDN and DMN regions which task set was activated and which task rule was applied.

The present univariate analysis results of the fMRI data are largely in accord with many previous findings. When a task cue is presented, the cue should be encoded and interpreted, leading to the establishment of the appropriate task set. While these processes are going on, a number of brain regions on the fronto-parietal cortex were found to be activated [2,20,30,31,32]. Indeed, a set of such regions, named the “Multiple Demand network”, were commonly activated by a variety of cognitive tasks and the activity of the MD regions proportionally increased as the demand of the task increased [2,3,33].

More recent studies regarding the functional role of the MD network showed that this network was not only sensitive to task demands or the amount of processing resources recruited, but also represented specific task features. The present study also largely replicated these findings; in the face of a task cue and a multi-dimensional target stimulus, activation patterns of the core MD regions discriminated which dimension of the stimulus was attended and which task set was active. Specifically, we examined whether the currently activated task set could be decoded from activation patterns evoked by the cue stimulus and the target stimulus. Notably, while the target-evoked activity reliably informed which task rule was being applied across the two experiments in the present study, rule decoding by the cue activity was significant only when the cue–target SOA was relatively short. Specifically, when the cue and the target stimulus were separated by 0.5 s, 1.5 s, 2.5 s, or 7.5 s, activation patterns of the dACC, LPFC, and IPS in the MDN, as well as those of the TPJ and pCC in the DMN, discriminated which task set (face or place task) was relevant. Notably, the present MDN pattern analysis results are also a replication of a well-established finding in the literature [17,34]. However, with longer delays, ranging from 4 s to 12 s, the cue-evoked activity did not inform which task set was relevant. There are several possible explanations for these results. First, with the short cue–target intervals, the demand for processing the cue and implementing the appropriate task set might increase, while, with the long interval, the cognitive load of encoding the cue and maintaining one of the two simple rules might have been trivial. With increased cognitive load under rapid timing, sharper task representations might have been needed [4]. Second, it might be simply due to the fact that the cue-evoked activity and the target-evoked activity were not well separated. However, regarding this, it is important to note that, in Experiment 2, not only cue decoding, but target decoding was also enhanced. Hence, under short cue–target intervals, the demand for encoding task-specific features might really have increased. Further research to elucidate this issue would be fruitful.

Along with the cue-evoked activity, the target-evoked activation patterns of the present ROIs also discriminated which task set was relevant, especially in Experiment 2. These regions included the LPFC and IPS in the MDN, along with the PCC, mPFC, and TPJ in the DMN. This finding is worth discussing because, in terms of the univariate activities of the regions, the two networks showed contrasting patterns; the MDN showed positive activity, whereas the DMN showed deactivation under baseline. Deactivation of the DMN under cognitively demanding tasks stimulated a large body of studies addressing the issue of the DMN’s functional roles. Some studies attributed this finding to the possibility that the DMN has a functional role associated with high-level reasoning, semantic processing, or social interaction, distinct from those of the MDN [6,15,35,36]. However, other recent studies suggested that the DMN complements the MDN in cognitive control [37,38,39]. In such a study, during the performance of attention-demanding tasks, which deactivates the DMN, the functional connectivity between the DMN and MDN increased [39]. In another study, while the DMN activity level was inversely correlated with working memory load, the DMN activation pattern represented detailed memory contents [37].

Expanding these studies, we demonstrated that the spatial pattern of the DMN activation contained important information regarding the task rule. What the present study adds to the current volume of relevant research is that the DMN was involved in cognitive control for task performance, which involved neither semantic judgment, social processing, nor long-term memory retrieval [38,39]. Another notable point is that we employed a trial-by-trial control design, differently from previous block-design studies [37,39] and found that the DMN was related to dynamic adjustment of task sets in a trial-by-trial manner. Indeed, a previous study employing a similar paradigm also reported similar findings [19]. A novel contribution of the present study is that the ROIs of the DMN were verified to be deactivated during an attention-demanding task and these regions encoded task features. This provides direct evidence that the DMN encodes task features when cognitive control should be implemented in a trial-by-trial manner.

Considering the extant evidence and the present results, we argue that the DMN and MDN co-operate for successful performance of a cognitive task. To achieve the task goal, an attentional set should be properly established, which might be primarily related to the MDN activity [3,20]. Concurrently with this, the meaning, significance, and behavioral relevance of a given sensory input should be properly identified, interpreted, and evaluated [40]. Such evaluative processes might be commonly mediated by the MDN and the DMN [4,18,40,41,42]. Following or along with this evaluative process, the MDN might play a role in recruiting top-down processing resources to meet the task demands [18,40,43]. We suggest that it would be valuable to test this kind of specific hypotheses to clarify the functional roles of the DMN and MDN.

Finally, there are several limitations in the present study. First, as an anonymous reviewer suggested, it has to be verified that the present results obtained using the face/building stimuli would be generalized to other kinds of paradigms. We chose these stimuli because these two kinds of stimuli are known to activated distinct brain regions, opening up several future studies regarding this issue. In a future study, it would be fruitful to investigate the interplay between the fronto-parietal network and perceptual regions. Second, the issue of the cue–target SOA needs a thorough test. We happened to find that the cue–target SOA modulates cue decoding performance. This has to be evaluated with an independent testing.

To conclude, we provide evidence that the MDN and DMN co-operate and contribute to a transient and rapid adjustment of an attentional set to meet the task demand at hand. By functionally localizing each network and examining activation patterns of the two networks, we found that these networks commonly encoded task features, facilitating to produce adaptive behaviors. We suggest that the DMN has a role in analyzing and deciphering the functional meaning of an incoming stimulus and closely co-operates with the DMN, which is involved in recruiting attentional resources to meet the task demand at hand.

## Figures and Tables

**Figure 1 brainsci-13-01247-f001:**
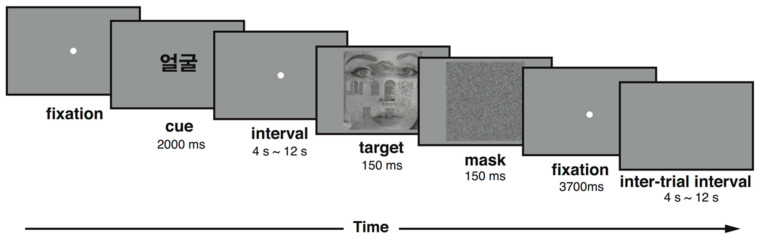
An example trial of Experimental 1.

**Figure 2 brainsci-13-01247-f002:**
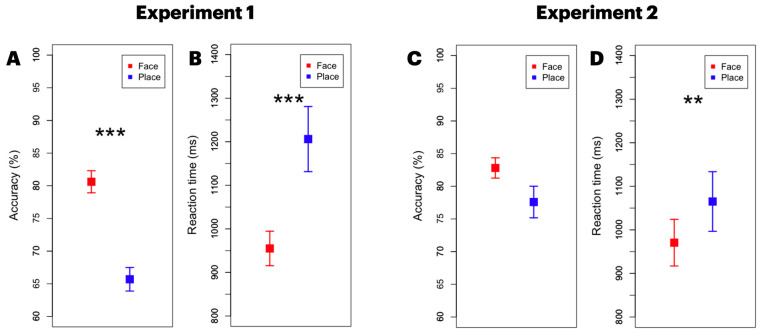
Behavioral results of both experiments. Accuracy results of Experiment 1 (**A**) and reaction time result (**B**). Accuracy results of Experiment 2 (**C**) and reaction time result (**D**). Error bars represent standard errors of mean. ** *p* < 0.01, *** *p* < 0.001.

**Figure 3 brainsci-13-01247-f003:**
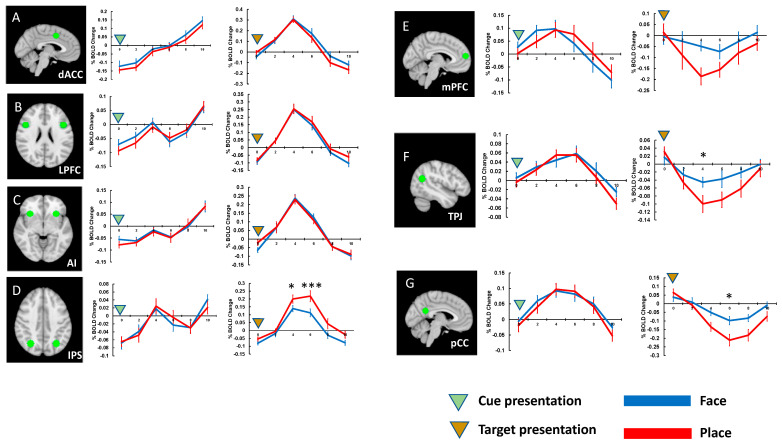
The locations of ROIs and activation timecourses (Experiment 1). (**A**) dACC and its activities to the cue and target presentations. (**B**) LPFC and its activities to the cue and target presentations. (**C**) AI and its activities to the cue and target presentations. (**D**) IPS and its activities to the cue and target presentations. (**E**) mPFC and its activities to the cue and target presentations. (**F**) TPJ and its activities to the cue and target presentations. (**G**) pCC and its activities to the cue and target presentations. Error bars represent standard errors of mean. * *p* < 0.05, ** *p* < 0.01, *** *p* < 0.001.

**Figure 4 brainsci-13-01247-f004:**
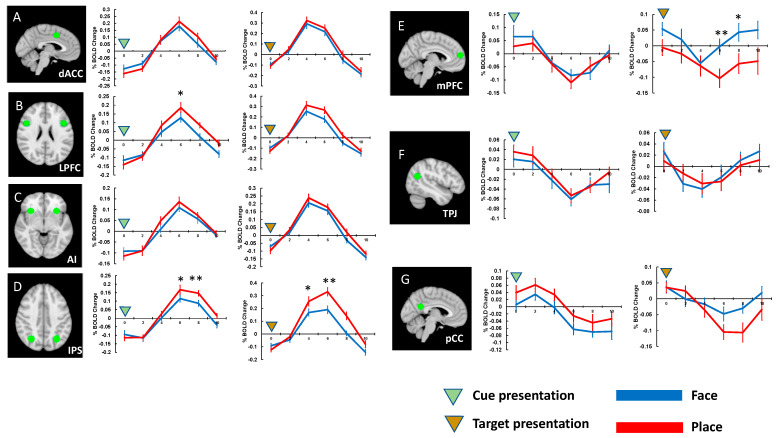
The locations of ROIs and activation timecourses (Experiment 2). (**A**) dACC and its activities to the cue and target presentations. (**B**) LPFC and its activities to the cue and target presentations. (**C**) AI and its activities to the cue and target presentations. (**D**) IPS and its activities to the cue and target presentations. (**E**) mPFC and its activities to the cue and target presentations. (**F**) TPJ and its activities to the cue and target presentations. (**G**) pCC and its activities to the cue and target presentations. Error bars represent standard errors of mean. * *p* < 0.05, ** *p* < 0.01, *** *p* < 0.001.

**Figure 5 brainsci-13-01247-f005:**
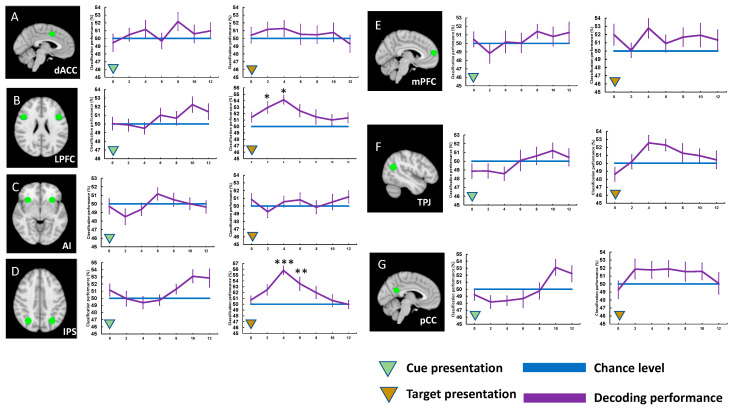
Multivariate pattern analysis (MVPA) results of Experiment 1. The locations of MDN regions (**A**–**D**) and DMN (**E**–**G**) regions are depicted. Error bars represent standard errors of mean. * *p* < 0.05, ** *p* < 0.01, *** *p* < 0.001, Bonferroni-corrected.

**Figure 6 brainsci-13-01247-f006:**
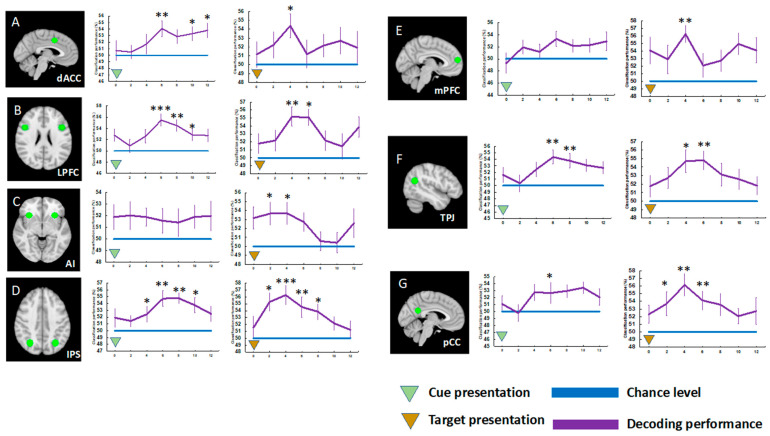
Multivariate pattern analysis (MVPA) results of Experiment 2. The locations of MDN regions (dACC, LPFC, AI, and IPS) (**A**–**D**) and DMN (mPFC, TPJ, and pCC) (**E**–**G**) regions are depicted. Error bars represent standard errors of mean. * *p* < 0.05, ** *p* < 0.01, *** *p* < 0.001, Bonferroni-corrected.

**Table 1 brainsci-13-01247-t001:** Co-ordinates of the regions of interest.

Networks	ROIs	X	Y	Z
MDN	dACC	−4	6	44
	LPFC			
	(left)	−44	12	24
	(right)	46	12	22
	AI			
	(left)	−30	22	−6
	(right)	32	22	−6
	IPS			
	(left)	−30	−64	40
	(right)	30	−64	40
DMN	mPFC	−6	58	14
	TPJ			
	(left)	−46	−60	22
	(right)	46	−60	22
	pCC	−4	−52	24

Co-ordinates are plotted in the MNI 152 space.

## Data Availability

All the data and codes will be publicly available on the acceptance of the paper; no part of the study procedures was preregistered prior to the research being conducted.; no part of the study analyses was preregistered prior to the research being conducted.

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
