# Peer review of "The Involvement of the Multiple Demand and Default Mode Networks in a Trial-by-Trial Cognitive Control"

_brainsci, 2023, doi:10.3390/brainsci13091247_

Round 1

Reviewer 1 Report

Dear authors,

The article deals with a very interesting issue,

and the work is original and appropriate.

Some methodological advice:

The anatomical images lack the technical parameters in the text.

Please rewrite and check the text, there is a lot of language problem. In some parts the reading is very tiring, with repetition of sentences and inappropriate grammar.

Author Response

The parameters of anatomical image acquisition is now added (FOV: 256 x 256, slice thickness: 1mm with no gap, in plane resolution: 1mm x 1mm, TR: 8.93 ms).

We also thoroughly proofread the paper and thank the reviewer for raising this issue.

Reviewer 2 Report

In the manuscript entitled “The involvement of the multiple demand and default mode networks in a trial-by-trial cognitive control”, authors analyzed two distinct sets of neuroimaging data, encompassing a total of 40 human participants. They observed that attentional task led to positive activity in the MDN and a reduction in activity within the DMN. Additionally, they reported that even the regions associated with the DMN exhibited the capacity to encode task-related features when rapid reconfiguration of the task was necessitated on a trial-by-trial basis, in tandem with the MDN regions. Authors speculate a collaborative interplay between these two distinct brain networks, hypothesizing their collective role in effectively establishing top-down cognitive control mechanisms.  

This is a well-conceived and interesting study.  However, in my view the manuscript is not ready for publication yet.

Comments:

1.      Overall, the manuscript would greatly benefit from a thorough evaluation and discussion of the recent literature. It is crucial for the authors to conduct a comprehensive evaluation and discussion of the recent literature on neural correlates of cognitive control. The current oversight of this aspect, despite its significance and alignment with the reported results (especially with the model proposed inhibitory demand by Gavazzi et al., 2021-23), hampers the manuscript's quality. The authors should focus on improving both the introduction and interpretation of their findings, particularly with regard to insula, dACC and prefrontal cortex. Incorporating and leveraging the following highly relevant and up-to-date papers will be essential for enriching the study:

Apsvalka, D., Ferreira, C.S., Schmitz, T.W., Rowe, J.B., Anderson, M.C., 2022. Dynamic targeting enables domain-general inhibitory control over action and thought by the prefrontal cortex. Nat. Commun. 13.

Choo, Y., Matzke, D., Bowren Jr, M.D., Tranel, D., Wessel, J.R., 2022. Right inferior frontal gyrus damage is associated with impaired initiation of inhibitory control, but not its implementation. Elife 11, e79667.

Gavazzi, G., Giovannelli, F., Currò, T., Mascalchi, M., & Viggiano, M. P. (2021). Contiguity of proactive and reactive inhibitory brain areas: A cognitive model based on ALE meta-analyses. Brain Imaging and Behavior, 15, 2199-2214.

Gavazzi G, Giovannelli F, Noferini C, Cincotta M, Cavaliere C, Salvatore M, Mascalchi M, Viggiano MP. (2023). Subregional prefrontal cortex recruitment as a function of inhibitory demand: an fMRI metanalysis. Neurosci Biobehav Rev.

2. 2.      Please include in the caption of the figures the name of the brain regions.

3.     3.   I believe that the employment of face stimuli may affect the generalizability of authors’ results. Not so much can be done at this point, however why you did not employ some neutral stimuli? Authors should justify this choice or at least include it in the limitation section of discussion.

4.       4. The discussion section of the paper needs a limitation and a future perspective section that in the this version can only be inferred by the main text.   

The manuscript doesn't have major English issues.

Author Response

Most importantly, we welcome the reviewer's suggestion that we need to mention the limitations of the present study and future directions. 

There are several limitations in the present study. First, it has to be verified that the present results obtained using the face/building stimuli would be generalized to other kinds of paradigms. We chose these stimuli because these two kinds of stimuli are known to activated distinct brain regions, opening up several future studies regarding this issue. In a future study, it would be fruitful to investigate the interplay between the fronto-parietal network and perceptual regions. Second, the issue of the cue-target SOA needs a thorough test. We happened to find that the cue-target SOA modulate cue decoding performance. This has to be evaluated with an independent testing.

We also appreciate the reviewer's suggestion regarding the relevant studies. We will certainly keep in mind that these are crucial for developing future studies. However, respectfully, we like to argue that these are not directly related to the present study because we are focusing on the MDN and DMN as networks, rather than individual brain regions. Furthermore, we are not sure how the present results are related to inhibitory control.

We also included the brain region names in the figure captions.

Reviewer 3 Report

A very technical and good approach on default mode 2 networks in a trial-by-trial cognitive control.

Study is of interest for those working in this area and

for some readers of the Brain Sci journal

introduction is very good

Methods are very complex and good.

Still, how do you define normal in visual acuity? did you test the subjects initially? it was just a preliminary screening. Just an interview?

Results are exhaustively presented, but as said, could be of interest for those working in this area.

Discussion is constructed very well.

Conclusions are short and balanced as it should be.

Good job!

Author Response

We appreciate the reviewer's comments and compliments.

We are happy to see that the reviewer caught the points we wanted to emphasize.

Regarding the participants' visual acuity issue, we had participants practice the task insider scanner, and if participants could clearly see the stimuli, we treated them to have normal visual acuity.